# Gender discrimination in hiring: An experimental reexamination of the Swedish case

**Ali Ahmed**[1], **Mark Granberg**[1]*, **Shantanu Khanna**[2]

**1** Division of Economics, Department of Management and Engineering, Linköping University, Linköping, Sweden, **2** Department of Economics, University of California Irvine, Irvine, California, United States of America

ʘ These authors contributed equally to this work.

* mark.granberg@liu.se

**Data Availability Statement:** All Data and Stata do-files are available from the OSF database (https://dx.doi.org/10.17605/OSF.IO/PJRF5).

**Funding:** AA and MG were financially supported by the Swedish Research Council (https://www.vr.se/) (grant no. 2018-03487).

## Abstract

We estimated the degree of gender discrimination in Sweden across occupations using a correspondence study design. Our analysis of employer responses to more than 3,200 fictitious job applications across 15 occupations revealed that overall positive employer response rates were higher for women than men by almost 5 percentage points. We found that this gap was driven by employer responses in female-dominated occupations. Male applicants were about half as likely as female applicants to receive a positive employer response in female-dominated occupations. For male-dominated and mixed occupations we found no significant differences in positive employer responses between male and female applicants.

## Introduction

Large differences between the occupational structures of men and women exist, even in developed and progressive countries. Occupational gender segregation is a persistent feature of labor markets, and accounts for a considerable portion of the wage gap [1]. According to the Global Gender Gap Report 2020 published by the World Economic Forum [2], Sweden ranks fourth of 153 countries on the Global Gender Gap index. Also, data from Statistics Sweden show that labor force participation rates for women in 2019 stood at 82.3 percent, only slightly lower than that of men at 85.9 percent. Even in Sweden, however, the gender wage gap persists, with estimates of the unadjusted gender gap in the range of 12 to 20 percent [3, 4]. Differences in occupation structures are estimated to account for more than half of this wage gap [5]. On the one hand, occupations like vehicle mechanics and truck drivers are overwhelmingly male, whereas preschool teachers and nurses are overwhelmingly female. Hiring discrimination is one potential channel that can limit progress on the path to more equal gender distributions within occupations and more equal wages.

Since Bertrand and Mullainathan [6] popularized the use of correspondence testing, the method has been widely used to study hiring discrimination for a variety of immutable

**Competing interests:** The authors have declared that no competing interests exist.

characteristics, such as race or gender, across several countries. In a correspondence test, fictitious applications are created and sent to real employers to study differences in positive employer response rates. While most correspondence studies have focused on race or nationality, usually finding evidence of discrimination against racial and ethnic minorities, there are far fewer studies on gender discrimination and results are mixed (see [7–10] for literature reviews). A recent comprehensive review by Baert [10] lists only eight [11–18] published correspondence studies on gender discrimination in hiring since 2005, in contrast to 34 studies on race and ethnicity. Some more recent work on gender discrimination not covered in Baert's review, e.g., [19–21], can be found in his online register of correspondence experiments on hiring discrimination (https://users.ugent.be/~sbaert/research_register.htm), but they are still few. Since labor markets are evolving continuously when it comes to female labor force participation rates, gender distributions in occupations, and gender wage gaps, complementary studies on hiring discrimination are necessary. Understanding gender discrimination in hiring, whether it stems from animosity (i.e., is taste-based) [22] or uncertainties related to workers' productivity (i.e., is statistical) [23, 24], and how it varies across occupations is important in order to develop policies that rectify gender-based disparities in the labor market.

In this study, we combine data from three previous studies to assess gender discrimination in hiring in Sweden. While these studies were originally designed to measure discrimination against other groups, their design allows for testing gender discrimination as well. Using data from over 3,200 fictitious job applications sent to employers across 15 occupations, we found that women had higher positive employer response rates than men on average (around 5 percentage points), with this difference being driven by female-dominated occupations (around 14 percentage points).

The heterogenous effects across occupational categories revealed that discrimination in hiring against men was much higher in female-dominated occupations. Overall, we found that in some male-dominated occupations like vehicle mechanics, warehouse workers and business-to-business (B2B) sales, men were likelier to receive positive employer responses than women, although these differences were not statistically significant. In female-dominated occupations such as cleaner, childcare provider, preschool teacher, accounting clerk, and enrolled nurse, positive employer response rates were much higher for women than for men. This is in line with earlier findings in different countries. In a meta-analysis of hiring discrimination studies during 2000–2014, Rich [9] documented high levels of discrimination against men applying to jobs in female-dominated occupations in China [13], France [14], England [25], and Australia [11].

Earlier reviews [26] have also noted that discrimination against men in female-dominated occupations is much higher than discrimination against women in male-dominated occupations. Indeed, when we analyzed our data according to male-dominated, mixed, and female-dominated occupations, we found an absence of statistically significant differences in positive employer response rates between men and women for the former two categories, but a large and highly significant male disadvantage in female-dominated occupations.

The most closely related prior studies to ours are Carlsson [16] and Carlsson and Eriksson [27] who also studied gender discrimination in hiring in Sweden. We tested for hiring discrimination across more occupational categories than these two studies or any other previous study that we know of, spanning a wider range of occupational gender shares. In addition, while these earlier studies used a matched pair testing (within-subjects) design, an approach which has come under criticism [28, 29], we sent a single application to each employer (between-subjects design) which is not vulnerable to these critiques.

Carlsson [16] concluded that the relationship between gender bias and female share across occupations is quite weak, and that women are "somewhat" more likely than men to receive

positive employer responses to their job applications in female-dominated occupations—that the correlation between the difference in positive employer response rates by gender and the share of women in the occupation is not significant. In contrast, we found that the difference in positive employer response rates between men and women is positively correlated with the share of women in the occupation, with a statistically significant correlation coefficient of .78. The estimates in [16] also differ from ours for some occupation categories. For instance, the differences in positive employer response rates were not statistically significant in that study for preschool teachers, cleaners, and nurses, whereas we found significant differences for these categories in our study. Carlsson [16] also ruled out in-group favoritism by finding no strong relationship between the relative treatment of female candidates and the gender of the recruiter or the proportion female at the firm. In contrast, Carlsson and Eriksson [27] *did* find evidence of in-group bias using very similar metrics. This raises salient questions that demand further exploration even within the context of Sweden.

## Experimental method and data

We combine the data from three previous studies where correspondence tests were conducted. All of these data are anonymized, i.e., individual firms, companies, or organizations are not identifiable. Moreover, these data were collected in Sweden where the Ethical Review Act [30] does not require ethical vetting of research involving firms, companies, and organizations. The data for Studies 1 and 2 consists of two separate samples which were collected during the same time period in the spring of 2016, while the data for Study 3 was collected during the spring of 2019. In this section, we describe the relevant aspects of these three data sets and the process of combining them. For more details on the individual studies, and their associated data collection, see [31] for Study 1, [32] for Study 2, and [33] for Study 3.

Correspondence tests are field experiments where the first stages of hiring is studied by sending out fictitious applications to real employers with available jobs. Employers' responses to the applications are then recorded for analysis. When the researchers' goal is to measure hiring discrimination, the group belonging of the fictitious applicants is experimentally manipulated, most commonly by randomizing the applicant's name. This method is preferable to alternative approaches (e.g., Kitagawa-Oaxaca-Blinder decomposition [34–36]), primarily because the experimentally induced variation in group belonging allows researchers to make causal claims regarding hiring discrimination. Of course, as with any method, there are drawbacks. Correspondence tests can naturally only measure discrimination at the initial stage of the hiring process and may understate the true extent of hiring discrimination if it occurs at later stages of the process. The method is therefore most suited for detecting the existence of discrimination and not its extent [37]. Other limitations include the difficulty of discerning between taste-based and statistical motivations behind discrimination and the inability of observing the competing applicant pool.

As pointed out by Phillips [28], another limitation that these studies can suffer from is due to a particular design choice. Namely that many researchers chose to run matched experiments, where multiple applications (often two or four) are sent to the same employer. While there are some appealing sides of this approach, there are also some problems. Phillips identified the potential risk of covariates of one applicant affecting the outcome of another. However, more serious in our view, is the risk that a matched design does not fulfill the Stable Unit Treatment Value Assumption (SUTVA). SUTVA is the assumption that the treatment of any one individual does not affect any other individual's outcome, and as Lewbel [38] points out: "When SUTVA is violated most causal inference estimators become invalid, and point identification of causal effects becomes far more difficult to obtain" (p. 866). Thus, a matched design

in correspondence tests may forgo or limit the stated goal and primary benefit of the method —the ability to arrive at causal estimates.

To further illustrate the problem related to SUTVA, consider the following example, set in the domain of our current inquiry: Imagine an employer looking to hire an enrolled nurse, the occupation in our sample which is most female-dominated with around 90 percent female workforce. The fewer applications an employer receives, the likelier it is that all of the job applicants will be women (so it would be true that these concerns are lessened in large and active labor markets). With an unmatched design, all of these differently sized applicant pools can happen to be studied by the experimenter and each single observation faces realistic competition as a result. However, if a matched pair (one male and one female applicant) is sent to each vacancy, then those observed female applicants are special in the sense that they always compete with a male applicant. It is not possible to know ex ante how this may bias the estimates of discrimination, but if the hypothetical employer has a will to diversify their workforce, this will negatively impact the positive employer response rates for observed female applicants making them an unrealistic control group. This would bias the estimates of discrimination against men in female-dominated occupations downwards. The inverse of this reasoning would similarly apply in male-dominated occupations.

It may seem ironic, with the views espoused in the preceding sections, that Studies 1 and 2 used in this paper were originally designed as matched experiments. However, these design choices were made before Phillips' [28] recent critique of the use of matched pairs in correspondence tests which moved our thinking on this issue. Fortunately, for this study we only needed to use the control applicants from Studies 1 and 2 and were able to discard the treatment observation from each testing pair. This was possible because gender was randomized across testing pairs in these studies. Because each observation was unpaired after dropping the treatment observations, they did not face artificial opposite gender competition and their treatment assignment did not interact in a way that would violate SUTVA. There was a potential level shift in that some observations faced competition from the treatment in Study 1 and some faced competition from the treatment in Study 2, but this can be controlled for by including dummy controls for each study. Study 3 featured an unpaired design from the outset.

When collecting the data for Study 1, the aim was to examine whether individuals who had been convicted of a crime were less likely to receive a positive employer response to their job applications. The focus of Study 2 was victims of crimes. Other than these differences in aims, the data collection and study designs were very similar, so in this paper we will often refer to these two samples jointly as Studies 1 and 2. It is important to note that the two samples collected during Studies 1 and 2 did not interact. Each day the research assistant flipped a virtual coin to decide for which sample they would be collecting data during that day. Because only job ads posted the previous day were applied to, this meant that Studies 1 and 2 randomly targeted different job postings. Twelve occupations were targeted with a good mix of male- and female-dominated sectors. In total, the data from Studies 1 and 2 contained 2,183 independent observations after we combined the crime victim and crime offender data sets and discarded the criminal or victim (i.e., treatment) observation from each testing pair.

Study 3 was conducted three years after Studies 1 and 2 in 2019 but at the same time of year. The aim was to examine whether transgender applicants faced hiring discrimination in the labor market. In contrast to the first two studies, Study 3 implemented a fully randomized vector of four skill variables, allowing for some more detailed analysis of how skills interact with the positive employer response rates for each type of applicant. After discarding the transgender applicant (i.e., treatment) observations, the data from Study 3 contained 1,071

independent observations with fully randomized assignment of gendered names (i.e., the desired one application per vacant job).

The age of applicants is a potential confounding variable when examining gender discrimination [39], in all three studies all applicants claimed to be 28 years old, something we likely would have varied, had the experiments been designed to test for gender discrimination at the outset. Altogether, the three studies encompassed 15 occupations, whereof eight were included in all three. These eight occupations were store clerk, vehicle mechanic, cleaner, enrolled nurse, waitstaff, chef, truck/delivery driver, and warehouse worker. Four occupations were not included in Study 3: preschool teacher, IT developer, B2B sales, and accounting clerk. Lastly, three occupations were unique to Study 3: customer service, telemarketing, and childcare.

For all three studies, occupations were chosen to cover a wide range of occupational gender ratios, and so there were enough vacancies posted frequently in each occupation to facilitate statistical tests. Gender ratio in occupation was hypothesized ex ante to interact with the original group signals (i.e. that men and women are likely to treat criminals, crime victims, and transgender people differently) so this was an explicit part of the design in all three studies. One job category, forklift operator, contained too few observations to facilitate statistical tests. Our solution was to combine warehouse worker and forklift operator into one category because they were similar by all measures, had been applied to using applications that were very similar, and exhibited similar response rates. The data are summarized in Table 1.

We classified occupations as female dominated if the share of women in the occupation exceeded 2/3 and as male dominated if that share was below 1/3; the rest were classified as mixed occupations (gender ratio data was acquired from Statistics Sweden's public database).

**Table 1. Summary statistics, rows sorted by occupational gender ratio.**

| Occupation | Gender ratio | | Observations | | | Positive employer responses | | | |
| | 2016 | 2019 | Studies 1 & 2 | Study 3 | Total | Total | Female applicant | Male applicant | Difference |
|---|---|---|---|---|---|---|---|---|---|
| Male dominated | (4.5) | (4.8) | 138 | 76 | 214 | 66 (30.8) | 25 (25.0) | 41 (36.0) | 16 (11.0) |
| Vehicle mechanic | | | | | | | | | |
| Delivery/truck driver | (8.3) | (8.7) | 222 | 115 | 337 | 116 (34.4) | 58 (34.1) | 58 (34.7) | 0 (0.6) |
| IT developer | (21.9) | - | 153 | 0 | 153 | 62 (40.5) | 33 (41.8) | 29 (39.2) | -4 (-2.6) |
| Warehouse worker | (22.3) | (22.9) | 29 | 112 | 141 | 27 (19.1) | 9 (14.3) | 18 (23.1) | 9 (8.8) |
| Mixed | (34.0) | - | 152 | 0 | 152 | 50 (32.9) | 21 (32.8) | 29 (33.0) | 8 (0.1) |
| B2B sales | | | | | | | | | |
| Telemarketing | - | (42.3) | 0 | 43 | 43 | 28 (65.1) | 13 (72.2) | 15 (60.0) | 2 (-12.2) |
| Chef | (53.1) | (50.9) | 223 | 169 | 392 | 99 (25.3) | 45 (25.0) | 54 (25.5) | 9 (0.5) |
| Waitstaff | (60.7) | (59.0) | 353 | 144 | 497 | 164 (33.0) | 86 (34.7) | 78 (31.3) | -8 (-3.4) |
| Store clerk | (62.2) | (61.9) | 27 | 100 | 127 | 23 (18.1) | 12 (19.0) | 11 (17.2) | -1 (-1.9) |
| Female dominated | - | (68.9) | 0 | 51 | 51 | 21 (41.2) | 13 (48.1) | 8 (33.3) | -5 (-14.8) |
| Customer service | | | | | | | | | |
| Cleaner | (75.2) | (74.0) | 295 | 139 | 434 | 88 (20.3) | 63 (27.8) | 25 (12.1) | -38 (-15.7)*** |
| Childcare | - | (81.9) | 0 | 71 | 71 | 28 (39.4) | 19 (52.8) | 9 (25.7) | -10 (-27.1)** |
| Accounting clerk | (82.1) | - | 166 | 0 | 166 | 38 (22.9) | 25 (27.5) | 13 (17.3) | -12 (-10.1) |
| Preschool teacher | (82.9) | - | 270 | 0 | 270 | 117 (43.3) | 66 (48.9) | 51 (37.8) | -15 (-11.1)* |
| Enrolled nurse | (91.7) | (90.9) | 155 | 51 | 206 | 83 (40.3) | 49 (46.7) | 34 (33.7) | -15 (-13.0)* |
| Total | (52.9) | | 2183 | 1071 | 3254 | 1010 (31.0) | 537 (33.3) | 473 (28.5) | -64 (-4.7)*** |

Note: This table reports summary statistics with percentages in parenthesis.

*, **, ***, indicate rejection of the null hypothesis that positive employer response rates are independent of applicant gender at the 10, 5, and 1 percent significance levels, respectively. Fisher's exact tests were used to evaluate the null hypotheses.

The outcome variable was binary, coded as 1 if a fictitious job applicant received a non-automated positive employer response (i.e., a request for more information about the applicant, an invitation for an interview, or an immediate job offer).

Beyond using Fisher's exact tests in Table 1, we used linear probability models (LPM) in our main analysis to estimate the rates of discrimination while controlling for covariates. As the primary outcome variable of interest was a positive employer response dummy, and neither zeroes nor ones were sparse in the data (mean positive response rate was 31 percent as seen in Table 1), LPMs were an appropriate and simple modelling choice. In the analysis we also controlled for the following covariates (often referred to as the following categories):

i.  Skill controls: Work experience (1–9), Squared work experience (1–81), Computer proficiency (0,1), English language proficiency (0,1), Active lifestyle (0,1).

ii.  Job controls: Full time position (0,1), Indefinite contract length (0,1), Job in urban area (0,1).

iii.  Study controls: Observation from Study 1 (0,1), Observation from Study 2 (0,1).

iv.  Occupation fixed effects (FE): a vector of dummy variables for each occupation with Chef as the reference category because this occupation had average positive employer response rates, was balanced in terms of gender ratio, and exhibited little differential treatment by applicant gender.

v.  Occupation level measures: Gender ratio in occupation (0–1, where 1 indicated 100 percent female workforce in occupation), Median occupational female-male wage difference in SEK (-5400–800, where negative values indicated a negative median wage difference for women). B2B sales was a substantial outlier in terms of median wage differences between men and women, these 152 observations were therefore excluded from some analyses where wage differences were a focus, in those cases the range of values for median occupational female-male wage difference in SEK was instead -1600–800. The USD to SEK exchange rate at time of writing (June 2020) was around 9.31.

The combined data that support the findings of this study are available at https://dx.doi.org/10.17605/OSF.IO/PJRF5.

## Results and analysis

Pooling data from all three data collections but without controlling for covariates the difference in mean positive employer response rates between male and female applicants was 4.74 percent in favor of women (bottom row of the difference column in Table 1). The difference was significant at the 1 percent significance level, $\chi^2(1, N = 3,254) = 8.52$, $p = .004$. For the rest of the analysis the outcome variable was the positive employer response dummy (see Tables A4 and A5 in S1 File for an analysis of robustness to different specifications of the outcome variable) and the independent variable of main interest was a dummy equal to one if the applicant was male. For details on covariates, see the Experimental method and Data section.

Table 2 shows the estimated marginal effect on positive employer responses from being a male applicant using linear probability models. Adding covariates had little impact on the main gender effect which remained stable at around 5 percent. As a balance of covariates test suggested that applicants' gender was successfully randomized (Table A1 in S1 File), this limited effect of adding covariates was to be expected.

Table 2 also shows that our skill variables––work experience, language skills, computer proficiency, and leading an active lifestyle––had a limited effect on positive employer response

**Table 2. Linear probability model marginal effects and covariates.**

| | (1) | (2) | (3) | (4) | (5) | (6) |
|---|---|---|---|---|---|---|
| **Treatment** | | | | | | |
| Male | -0.047*** | -0.047*** | -0.048*** | -0.050*** | -0.049*** | -0.050*** |
| | (0.016) | (0.016) | (0.016) | (0.016) | (0.016) | (0.016) |
| **Skills** | | | | | | |
| Experience | | 0.071*** | | | | 0.033 |
| | | (0.019) | | | | (0.025) |
| Experience$^2$ | | -0.007*** | | | | -0.002 |
| | | (0.002) | | | | (0.002) |
| Computer | | -0.072*** | | | | -0.024 |
| | | (0.028) | | | | (0.029) |
| Language | | -0.011 | | | | 0.030 |
| | | (0.027) | | | | (0.029) |
| Active | | -0.044 | | | | -0.012 |
| | | (0.027) | | | | (0.029) |
| **Job controls** | | | | | | |
| Fulltime | | | | 0.059*** | | 0.022 |
| | | | | (0.018) | | (0.019) |
| Contract Length | | | | -0.013 | | -0.052*** |
| | | | | (0.018) | | (0.018) |
| Urban | | | | 0.048*** | | 0.048*** |
| | | | | (0.016) | | (0.016) |
| **Study controls** | | | | | | |
| Study 1 | | | | | -0.130*** | -0.230*** |
| | | | | | (0.020) | (0.045) |
| Study 2 | | | | | -0.147*** | -0.244*** |
| | | | | | (0.020) | (0.045) |
| Constant | 0.334*** | 0.350*** | 0.282*** | 0.277*** | 0.429*** | 0.332*** |
| | (0.012) | (0.052) | (0.023) | (0.021) | (0.017) | (0.068) |
| Occupation FE | No | No | Yes | No | No | Yes |
| Observations | 3,254 | 3,254 | 3,254 | 3,252 | 3,254 | 3,252 |

Note: This table reports the marginal effect of being a male applicant using linear probability models with different sets of control variables. Robust standard errors in parentheses.

*, **, ***, indicate rejection of the null hypothesis at the 10, 5, and 1 percent significance levels, respectively. "FE" is short for fixed effects. Chef was the reference category for the occupation FE, the estimates for the occupation FEs have been omitted to save space. Study 3 was the reference category for the study controls.

rates once we accounted for occupation fixed effects (Column 6). The skill variables were non-randomized and fixed across occupations in Studies 1 and 2, the occupation fixed effects were therefore likely to have captured some of their effects in the full model. In the following analysis, the specification in column 6 was used unless otherwise specified.

As for heterogeneous effects across occupational categories, Table 3 shows the estimated marginal effect of being a male applicant in male-dominated, mixed, and female-dominated occupations for the combined sample and for each of the three studies separately.

Results from all three studies showed an overall penalty in positive employer response rates for men but the effect was not significant in Study 3. We found no evidence of discrimination in male-dominated and mixed occupations; the estimates were not significantly different from zero. In female-dominated occupations however, there was clear, consistent, and large

**Table 3. Marginal effects of being male on positive response rates in occupations with different gender ratios.**

| | (1) | (2) | (3) | (4) |
|---|---|---|---|---|
| | **All occupations** | **Male-dominated occupations** | **Mixed occupations** | **Female-dominated occupations** |
| All studies | | | | |
| Male | -0.050*** | 0.040 | -0.017 | -0.143*** |
| | (0.016) | (0.031) | (0.025) | (0.026) |
| Constant | 0.332*** | 0.573*** | 0.366*** | 0.299** |
| | (0.068) | (0.184) | (0.101) | (0.132) |
| Observations | 3,252 | 844 | 1,210 | 1,198 |
| Study 1 | | | | |
| Male | -0.067** | -0.015 | -0.055 | -0.108** |
| | (0.026) | (0.052) | (0.044) | (0.043) |
| Constant | 0.499*** | 0.523*** | 0.470*** | 0.158 |
| | (0.088) | (0.182) | (0.141) | (0.110) |
| Observations | 1,049 | 255 | 381 | 413 |
| Study 2 | | | | |
| Male | -0.048* | 0.064 | -0.013 | -0.138*** |
| | (0.025) | (0.054) | (0.042) | (0.039) |
| Constant | 0.433*** | 0.255* | 0.685*** | 0.043 |
| | (0.080) | (0.140) | (0.126) | (0.096) |
| Observations | 1,132 | 286 | 373 | 473 |
| Study 3 | | | | |
| Male | -0.034 | 0.060 | 0.012 | -0.204*** |
| | (0.029) | (0.054) | (0.045) | (0.055) |
| Constant | 0.451*** | 0.254 | 0.453*** | 0.435*** |
| | (0.081) | (0.157) | (0.114) | (0.157) |
| Observations | 1,071 | 303 | 456 | 312 |
| Occupation FE | Yes | Yes | Yes | Yes |
| Job controls | Yes | Yes | Yes | Yes |
| Skill controls | Yes | Yes | Yes | Yes |

Note: This table reports the marginal effect of being a male applicant in occupations with different gender ratios using linear probability models. Robust standard errors in parentheses.

*, **, ***, indicate rejection of the null hypothesis at the 10, 5, and 1 percent significance levels, respectively. Occupation fixed effects, skill controls, and job controls were included in each model, i.e., specifications in line with column 6 of Table 2.

discrimination against men. When using all data, the *p*-value for the negative marginal effect for men would have been considered significant in high energy physics ($p = .000000026$ or, to use physicist notation, $5.57\sigma$). Thus, using the combined sample, we estimated that female applicants had a 52.17 percent relative advantage in positive employer response rates over males in occupations where they were the predominant gender.

One possible caveat to consider is that the variance of unobservables can influence estimates of discrimination as Heckman and Siegelman proposed in their critique of audit studies [40]. We, therefore, used Neumark's method [41] to decompose the estimate of discrimination into one part which is due to this variance and one which is not. As the variance part of the decomposition was not significant in either case, both the full sample and the female-dominated occupation estimates were robust to the Heckman and Siegelman critique [40]. These results are presented in Table A2 in S1 File.

As gender ratio in occupations is a continuous variable which varies both by occupation and by year (i.e., by study) its interaction with the male treatment variable can be graphed. This is shown in Fig 1 where the interaction in a probit model are shown graphically (for the probit estimates underlying Fig 1 see Table A7 in S1 File where we also show that LPM estimates were similar). In Fig 1 we can see the negative relationship between male-female differences in positive employer response rates and gender ratio in occupation.

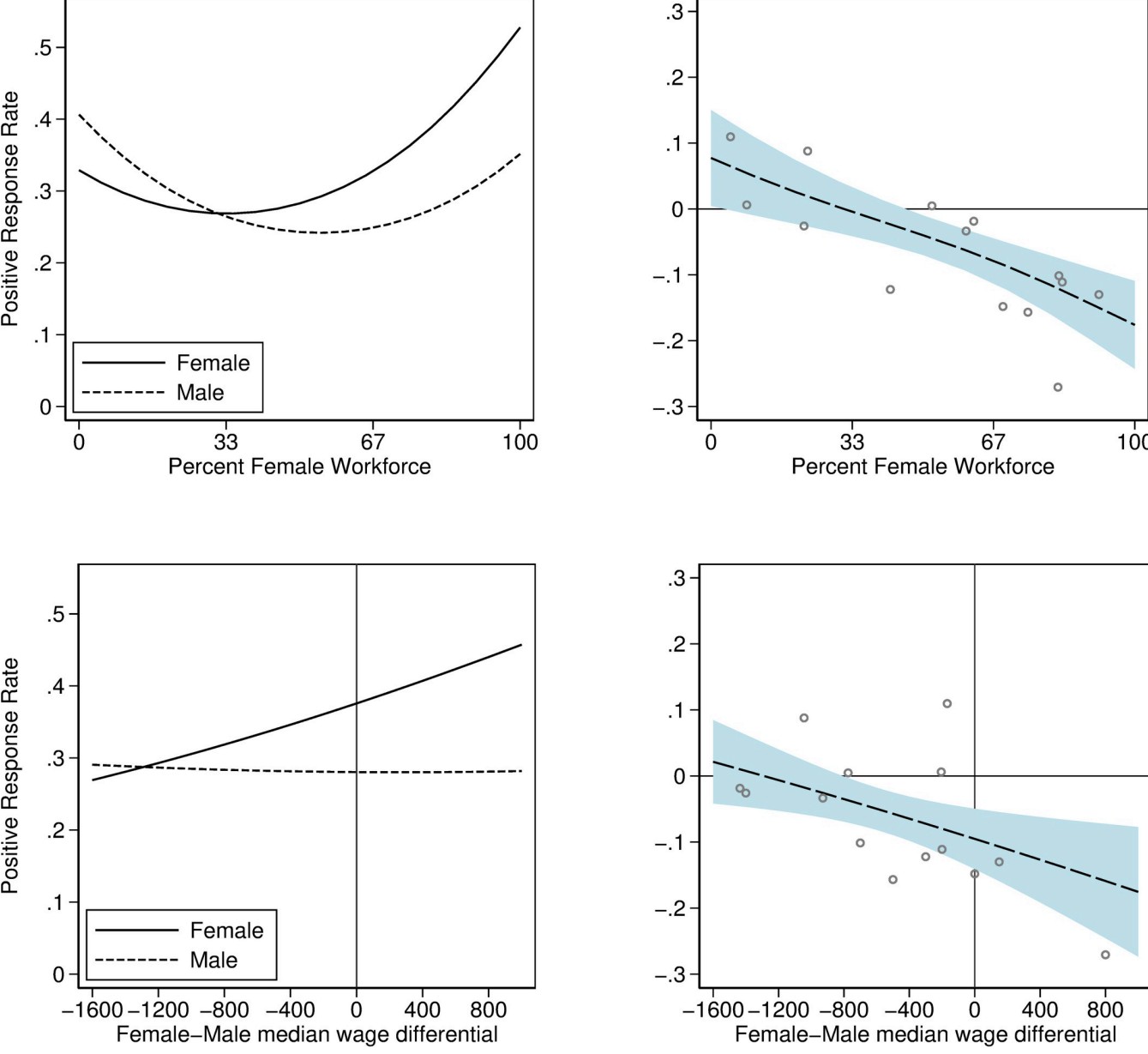

**Fig 1. Interaction between occupational characteristics and male applicant.** These graphs are based on a probit model including gender ratio and median wage difference interactions. The left-hand graphs show the predicted probability of a positive employer response for applicants given occupation level characteristics. The top two graphs show how occupational gender ratio interacts with the male treatment and the bottom two graphs show how occupational wage difference interacts with the male treatment. The right-hand graphs plot the differences in predicted probability of a positive employer response, and the circles indicate the raw mean differences in positive employer responses for each occupation. B2B sales is a substantial outlier in median wage differential and would blow out the x-axis scale in the bottom two graphs without adding much information, those 152 observations were therefore omitted from these regressions.

In Sweden, wage offers are essentially never posted in job ads, and there is usually an individual negotiation about wages for each new hire. It is therefore possible that employers may discriminate against men if they expect them to demand higher wages. In the second set of graphs in Fig 1 we used occupational median wage differences to illustrate that discrimination against males seemed largely insensitive to differences in wage expectations between the genders (for probit estimates, see Table A7 in S1 File). If anything, the differences in positive employer response rates run counter to expectations. Had these differences in positive employer response rates been due to differences in expected wages, one would expect to see a positive relationship shown in the bottom right panel of Fig 1, as the *x*-axis represents lower male wages going from left to right.

A common test of whether observed discrimination is statistical, is to examine if discrimination declines as observable skills increase. Unfortunately, there is never any guarantee that the stated skills in applications are the ones that interact with the independent variable of main interest [8]. For example, employers may discriminate against women or men differently based on the gender signal's implications for the likelihood of physical strength or fertility but may not make any assumptions about an applicant's language skills based on gender. In our case these studies were not designed to measure statistical discrimination against men or women, and as such productivity-relevant factors which can be reasonably argued to differ between the sexes were not experimentally manipulated (skills were not varied at all in Studies 1 and 2), which limits us in our analysis. Indeed, skills were only weakly associated with positive employer response rates in these studies (column 6 in Table 2), perhaps because most of the included occupations could be classified as low skilled. With these reservations in mind we still preformed the tests and found no strong evidence of statistical discrimination (see Fig A6, and the accompanying discussion in S1 File) which, of course, is not to say that discrimination is not statistically motivated on some factor which was not varied in our applications.

As for taste-based motivations they could potentially be intrinsic to the employer; the employer could be acting in accordance with the tastes of their customers; or the tastes of their employees [22]. As is typical for correspondence tests, distinguishing among these sources of tastes is challenging. The best we could do was to proxy customer tastes by how much customer interaction (CI) the occupations tend to require. We therefore classified store clerk, cleaner, enrolled nurse, customer service, waitstaff, telemarketing, preschool teacher, childcare, truck/delivery driver, and B2B sales as high CI occupations and the others as low CI occupations. When this CI indicator was interacted with the dummy variable for gender in an LPM we found some indication that discrimination against male applicants was driven by customers' tastes ($\beta = -.092$, $p = .006$, for full estimates see Table A3 in S1 File).

As for co-worker tastes, the only test available to us was to see if the dummy variable for gender interacted with whether the respondent was female or not, which restricted our sample, as data on respondents' gender was only collected in Studies 1 and 2. The hypothesis, based on the well-known in- and out-group group bias in social psychology, was that women would favor female applicants and men would favor male applicants. The interaction between the dummy variable for gender and female respondent, however, was not significant in an LPM specification and therefore provided us with no evidence of co-worker discrimination ($\beta = -.063$, $p = .158$, for full estimates see Table A3 in S1 File). Of course, it is possible that co-worker tastes are a partial explanation for the differences in discrimination across occupational gender ratios discussed above, but we are unable to identify this effect.

To summarize the results, we found that men were discriminated against on the whole in the three studies. This discrimination was most notably concentrated in female-dominated occupations, where men suffered a 14.3 percentage point penalty in positive employer response rates compared to women. In other words, women enjoyed a 52.2 percent relative

advantage over men in female-dominated occupations. We found no support for the statistical discrimination hypothesis, though this result may be due to limited power and weak relevance of the skill indicators used. Lastly, we found some indication that part of the discrimination could be attributed to customers' tastes.

## Discussion

Observing the discrimination against men in female-dominated occupations we have documented herein may lead one to wonder if these occupations have failed to integrate men. If we examine administrative data on occupational workforce compositions from Statistics Sweden, both male- and female-dominated occupations seem to be slowly integrating and, if anything, this process seems to have been swifter in female-dominated occupations.

The left-hand graph of Fig 2 shows that the workforce weighted mean share of women in both male- and female-dominated occupations has been slowly moving towards parity during the 21st century. The large gap in positive employer response probabilities we have documented would seem at odds with this occupation level convergence, unless there has been an increased inflow of males applying for female-dominated jobs, or if a disproportionate share of men actually end up with jobs *after* the initial stage of the hiring process. A natural shortcoming of correspondence studies is that we are unable to observe the applicant pool from which each employer was choosing. However, administrative data from the Swedish National Agency for Education on the rate at which men and women graduate from secondary school programs specifically aimed at these gender-imbalanced occupations could serve as an indicator for the general gender mix of applicants. The four gender-imbalanced occupations where our fictitious applicants reported a non-general education were enrolled nurses and childcare (female dominated) and vehicle mechanic and warehouse worker (male dominated), most other applicants reported three years of a general social science program. In the right-hand graph of Fig 2

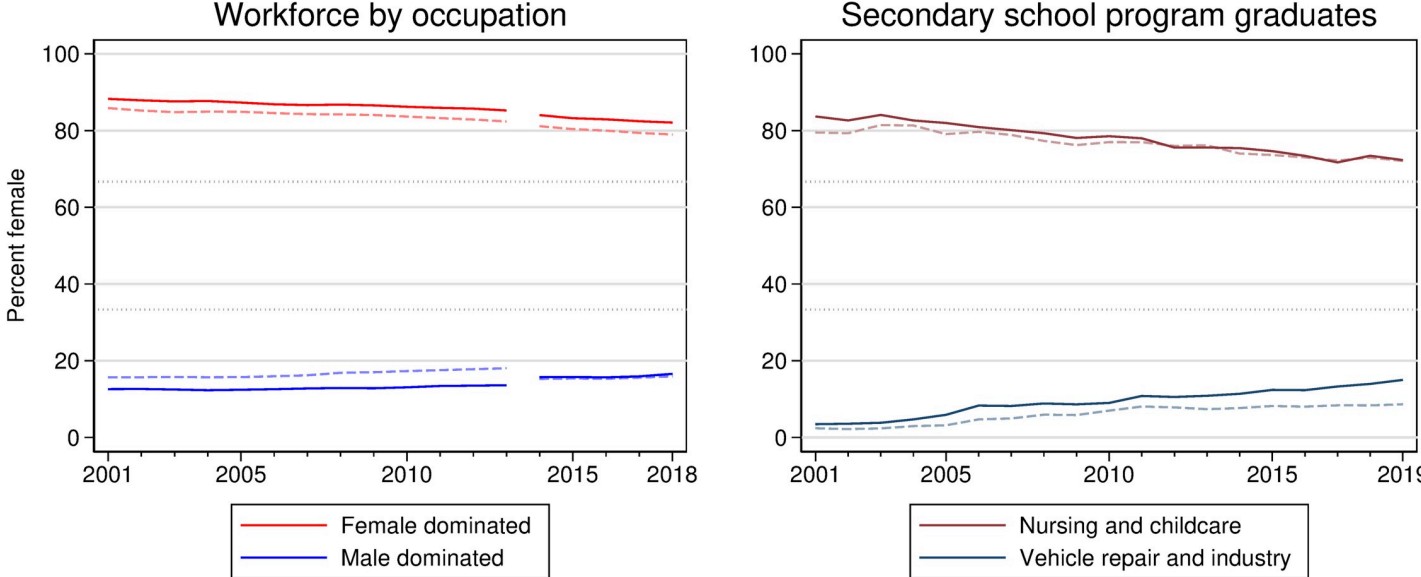

**Fig 2. Weighted mean gender ratios in male- and female-dominated occupations and secondary school programs.** Data from Statistics Sweden and the Swedish National Agency for Education. The left-hand graph shows the workforce weighted mean share of women in the male dominated (Blue) and female-dominated (Red) occupations included in this study. The dashed lines are the workforce weighted means for all occupations (SSYK 3-digit classification). Because there was a reclassification of occupational categories between 2013 and 2014 the first part of the series was classified as female (male) dominated if the share of women (men) was above two thirds in 2001 and this same classification was done for 2014 for the second part of the series. The right-hand graph shows the total graduates weighted share of women graduating from secondary school programs which were female-dominated (maroon) and male dominated (navy) in 2001. Solid lines are for the programs included in our studies (for most occupations applicants would report more general programs) while dashed lines are overall weighted averages.

we can see that there has been more rapid convergence in the gender mix of secondary school graduates in gender-imbalanced programs.

This increased supply of opposite-gender candidates may have had different effects in male- and female-dominated occupations. In the male-dominated occupations our results show that women are not discriminated against in hiring, perhaps due to increased awareness of such discrimination in society. In female-dominated occupations, if we consider the results of Carlsson's 2011 study [16] where data from 2005–2006 was used, there was little or no discrimination against men when fewer men were applying for these jobs. But as the supply of men has increased so too has the discrimination against them in hiring (this trend also holds in our data, but we find it less convincing with Studies 1 and 2 being so close temporally to Study 3). One possible explanation for this observation is that female-dominated workplaces could have been looking for some male representation but that once some men have been hired, the demand quickly diminishes and without the social pressure to balance the gender mix further, male applicants start experiencing discrimination.

Another more common explanation offered in the literature for lower positive employer response rates is gender stereotyping. The female-dominated occupations where we find significant discrimination involve considerable interaction with children (preschool teacher and childcare), with the ailing (nursing), or with families (cleaners). Research in psychology suggests that female stereotypes are associated with communality [42]. Manzi [43] points out how this can translate to discrimination against men in female-dominated occupations:

> "communality is perceived to be a requisite for success in traditionally female roles and occupations.... Congruity Models of Discrimination predict that the mismatch between people's perceptions of female-typed occupations and male stereotypes will lead to the belief that men will be less competent than women in these settings. These negative competence expectations should, in turn, lead to anti-male bias and discrimination against men in traditionally female domains."(43 p6)

Our findings on hiring discrimination against men in typically female domains are in direct agreement with Manzi's theoretical predictions. But the reasoning seems to leave the lack of discrimination in male-typed occupations unexplained.

From an economic point of view, discrimination may lead to inefficient market solutions and resource allocations. These labor markets may be failing to make optimal use of available human capital. There is an important distinction here between employer/co-worker tastes on the one hand and customer tastes on the other. It could be economically optimal to discriminate in hiring if it is in response to customer demand. Unfortunately, our results only give a vague indication that customer discrimination seems to matter, so more innovative methods to study the drivers of discrimination are clearly needed. However, no matter if discrimination originates from customers', co-workers', or employers' tastes or if it is based on statistical motivations, it is still a violation of human rights.

## Conclusion

Combining data from three previous correspondence studies in Sweden we found evidence of hiring discrimination against men on average. Examining responses to resumes in 15 different occupations allowed us to study how discrimination varied by occupations characterized by widely different gender ratios. Consistent with findings across several countries, but in contrast to some previous findings in Sweden [16], we observe high levels of discrimination against men in female-dominated occupations. Hiring discrimination is one demand-side

explanation for very skewed gender ratios in some occupations, which remains a barrier to gender equality.

## Supporting information

**S1 File. The complementary analyses referenced throughout this paper.**
(PDF)

## Acknowledgments

The authors thank Elisabeth Lång who carried out Studies 1 and 2 together with the first author, as well as Per A. Andersson who worked with us on Study 3. We also want to acknowledge the research assistance of Rickard Mobäck in the process of collecting the data for Study 3. Comments and suggestions from David Andersson, Per A. Andersson, Stijn Baert, Roger Bandick, Bart Capéau, Magnus Carlsson, David Neumark, Eva Van Belle were much appreciated and greatly improved this paper.

## Author Contributions

**Conceptualization:** Ali Ahmed, Mark Granberg, Shantanu Khanna.

**Data curation:** Mark Granberg.

**Formal analysis:** Mark Granberg, Shantanu Khanna.

**Funding acquisition:** Ali Ahmed.

**Investigation:** Mark Granberg, Shantanu Khanna.

**Methodology:** Ali Ahmed, Mark Granberg, Shantanu Khanna.

**Project administration:** Mark Granberg.

**Resources:** Ali Ahmed.

**Supervision:** Ali Ahmed, Mark Granberg.

**Visualization:** Mark Granberg, Shantanu Khanna.

**Writing – original draft:** Mark Granberg, Shantanu Khanna.

**Writing – review & editing:** Ali Ahmed, Mark Granberg, Shantanu Khanna.

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
