## [Decision Letter · Decision Letter 0]

4 Nov 2020

PONE-D-20-31057

Gender discrimination in hiring: An experimental reexamination of the Swedish case

PLOS ONE

Dear Dr. Granberg,

Thank you for submitting your manuscript to PLOS ONE. After careful consideration, we feel that it has merit but does not fully meet PLOS ONE’s publication criteria as it currently stands. Therefore, we invite you to submit a revised version of the manuscript that addresses the points raised during the review process.

I agree with the reviewers that this article is a sound contribution and that after revision work, in line with the reviewers' comments, there is a high probability that it will be suitable for publication in PLOS ONE.

From my side, I would ask you to acknowledge a bit more recent field experimental research (published the last five years) on hiring discrimination against females. My register can guide you in this respect: https://users.ugent.be/~sbaert/research_register.htm.

In addition, it would be great if you could propose a communication strategy to maximise the article's scientific ànd societal impact upon publication.

We look forward to receiving your revised manuscript.

Kind regards,

Stijn Baert, Ph.D.

Academic Editor

PLOS ONE

Reviewers' comments:

Reviewer's Responses to Questions

**Comments to the Author**

1. Is the manuscript technically sound, and do the data support the conclusions?

Reviewer #1: Partly

Reviewer #2: Yes

2. Has the statistical analysis been performed appropriately and rigorously? 

Reviewer #1: Yes

Reviewer #2: Yes

3. Have the authors made all data underlying the findings in their manuscript fully available?

Reviewer #1: Yes

Reviewer #2: Yes

4. Is the manuscript presented in an intelligible fashion and written in standard English?

Reviewer #1: Yes

Reviewer #2: Yes

5. Review Comments to the Author

Reviewer #1: I find the analysis in this paper neatly done. I have two problems though. First, in its present state, the choices made by the authors are poorly motivated. Second, too few information on the raw sample is given, and this is in my view indispensable for this type of research.

In my reviewer report in attachment, I give more details on these points.

Reviewer #2: The paper estimates gender discrimination in hiring combining data from three large-scale field experiments in Sweden. They find that men have lower hiring probabilities overall, and especially when applying for a female-dominated occupation.

While some evidence on gender based hiring discrimination exists even for Sweden, it is true that this evidence has not been totally conclusive. The correspondence experiments used in the paper are appropriate to measure this discrimination and seem well-designed and executed. Moreover, the paper is well-written. I nevertheless have some comments which I believe the authors should address.

1. The authors go very quickly over a number of key concepts, making the paper difficult to read for anyone not familiar with the literature or methodology used.

a. Apart from one sentence (p. 2 line 54), the authors do not explain what a correspondence experiment is, why it is an appropriate method to measure hiring discrimination, or what it’s main advantages and disadvantages are as opposed to other methods.

b. On p. 15, line 311, the authors talk about statistical discrimination without defining this theory. The same is true or taste-based discrimination further in the document.

2. Further on page 15, the authors state “In our case, skills were only weakly associated with positive employer rates in general.”. It is unclear from which analysis the authors draw this conclusion. This entire paragraph is confusion and the reader lacks information in order to understand the conclusions drawn here.

a. Moreover, as statistical discrimination occurs when employers make assumptions about a candidate’s skills following imperfect information, it is more about the quantity of the skill-related signals than the quality. This is something the authors cannot test for, but should be acknowledged.

3. The paper misses a “methodology” section. Parts of the method are explained in the “Data” section and others in the “Results” part, making the structure of the paper difficult to follow

6. PLOS authors have the option to publish the peer review history of their article (what does this mean?). If published, this will include your full peer review and any attached files.

Reviewer #1: **Yes: **Bart Capéau

Reviewer #2: **Yes: **Eva Van Belle

---

## [Author Response · Author response to Decision Letter 0]

16 Nov 2020

Response to Editor

We thank the editor for his fair and reasoned evaluation of our manuscript. In response to the first comment about references we have amended the section about previous studies to include both, as the editor suggested, more recent work but also added references to previous work which was previously lumped into “eight published correspondence studies on gender discrimination in hiring”. We have also included the suggested link to the correspondence test repository.

As for the communication strategy, here follows a number of steps we will take, should the paper be accepted:

1. We will work with our competent communications department here at Linköping University and inform them about our publication. They in turn will communicate our results through our home page, Twitter, Instagram, and Facebook. They typically also do a press release in both Swedish and English meaning that our results will be communicated to both national and international press outlets.

2. We will work to disseminate these results in our own research network, specifically within a national Swedish collaboration on research in labor market discrimination supported by the Swedish Research Council. 

3. Furthermore, we will share our work at seminars both at Linköping University and University of California, Irvine.

Response to Reviewer 1

Major points

1. We acknowledge that views differ on this point, and we appreciate the opportunity to expand on our reasoning around matched pair testing in correspondence tests. We have done so in the extended data section which is now titled “Experimental Method and Data” (see pages 6-8). We think our position on the matter is made clearer by the additions, and thank reviewer 1 for his input which, we think, greatly improved our argumentation on the issue in this paper.

2. We agree that this was not presented properly in the previous version and have clarified exactly which interaction is being referred to (see page 16). We now inform the reader that probit estimates are available in table A7 of the supplementary material (which we added). We have also added LPM-estimates to table A7 to show that they are similar to probit estimates. In going over this analysis we found a small error in our code, which marginally changed the look of our graphed interaction effects in this particular instance, though nothing of substance changed we thank reviewer 1 for prompting us to go over this analysis again to catch the minor error.

Minor points

1. Our intention in the introduction was to only briefly mention gender wage gaps in order to motivate the importance of occupational segregation, and the contribution of the latter to the former. We find it to be an important part of motivating why we should care about gender discrimination in the labor market, and to provide important contextual information. We have looked over our writing on this in the hopes of clarifying this view.

2. We have added a note in the method section on the age of applicants being 28 in all three studies. As for the other information requested, it is available in Table 1. The first panel titled gender ratio provides the share of female workers in each of the occupations for 2016 and 2019. As regards the balance of male and female CVs sent to each occupation, this share is calculable from the information in the table and is shown to not be an issue by the balance of covariates tests in Table A1 in the supplementary material. We are quite limited in what additional information would fit in Table 1, and therefore hope that this explanation will be sufficient.

3. We have expanded on this, hopefully it is a lot clearer now (Page 15-16).

4. Yes, hopefully this is clearer now as we have added some discussion about this in the remade method and data section.

5. Yes, this is very possible, and we now discuss this in the paper. We also further emphasize now that skills were only varied in study 3 in the additions on page 17.

Response to Reviewer 2

1. We have tried to expand on these concepts

a. We agree this was too sparsely explained in the submitted version and have expanded the description, especially in the extended data section which is now titled “Experimental Method and Data”. Specifically, the second paragraph in this section explains in greater detail what correspondence studies are, and how they are superior to other regression-based approaches to measuring discrimination (which are not causal). 

b. We agree statistical discrimination came up rather suddenly as presented in the submitted version. We therefore now briefly introduce the concept in the introduction; but as this analysis is not a focus of the paper, we feel that we do not want to take up too much space in the introduction by discussing well-established concepts in the literature on discrimination.

2. We have tried to clarify this section and made explicit reference to the specific table and column in which the weakness of our skill variables is shown.

a. We understand that it is not possible to satisfactorily determine whether statistical discrimination is more about the quantity of the skill-related signals as opposed to quality. The theory, as proposed, does not make this distinction, and we remain agnostic about that in the paper as well..

3. Yes, we agree this was not optimally executed. Our challenge was that really the “method” for this study is in the combining the different data sets and thus we focused on their similarities and differences. We have expanded our description of the experimental approach, and moved some of the method-parts away from the results section We feel that combining the experimental method and data sections is a reasonable compromise as describing the data is so tightly interconnected with the method. We hope that the revised section is clearer.

General Response

We thank the editor, Stijn Baert, and the two referees, Bart Capéau and Eva Van Belle, for their invaluable input. In our view, their comments and perspectives definitely helped us produce a significantly improved article.

---

## [Editor Report · Decision Letter 1]

2 Jan 2021

Gender discrimination in hiring: An experimental reexamination of the Swedish case

PONE-D-20-31057R1

Dear Dr. Granberg,

We’re pleased to inform you that your manuscript has been judged scientifically suitable for publication and will be formally accepted for publication once it meets all outstanding technical requirements.

Kind regards,

Stijn Baert, Ph.D.

Academic Editor

PLOS ONE
---

## [Editor Report · Acceptance letter]

11 Jan 2021

PONE-D-20-31057R1 

Gender discrimination in hiring: An experimental reexamination of the Swedish case 

Dear Dr. Granberg:

I'm pleased to inform you that your manuscript has been deemed suitable for publication in PLOS ONE. Congratulations! Your manuscript is now with our production department. 

Kind regards, 

on behalf of

Professor Stijn Baert 

Academic Editor

PLOS ONE